# Ground Validation and Error Sources Identification for GPM IMERG Product over the Southeast Coastal Regions of China

**Xinxin Sui** [1], **Zhi Li** [2], **Ziqiang Ma** [1,*], **Jintao Xu** [1], **Siyu Zhu** [1] **and Hui Liu** [1]

1   Institute of Remote Sensing and Geographical Information Systems, School of Earth and Space Sciences, Peking University, Beijing 100871, China; suixinxin95@163.com (X.S.); jintaox@zju.edu.cn (J.X.); zhusiyu2019@pku.edu.cn (S.Z.); huil@pku.edu.cn (H.L.)
2   School of Civil Engineering and Environmental Science, University of Oklahoma, Norman, OK 73072, USA; li1995@ou.edu
*   Correspondence: ziqma@pku.edu.cn; Tel.: +86-188-0653-4289

**Abstract:** The Integrated Multi-satellitE Retrievals for the Global Precipitation Measurement mission (IMERG) has been widely evaluated. However, most of these studies focus on the ultimate merged satellite-gauge precipitation estimate and neglect the valuable intermediate estimates which directly guide the improvement of the IMERG product. This research aims to identify the error sources of the latest IMERG version 6 by evaluating the intermediate and ultimate precipitation estimates, and further examine the influences of regional topography and surface type on these errors. Results show that among six passive microwave (PMW) sensors, the Microwave Humidity Sounder (MHS) has outstanding comprehensive behavior, and Special Sensor Microwave Imager/Sounder (SSMIS) operates advanced at precipitation detection, while the Sounder for Atmospheric Profiling of Humidity in the Intertropics by Radiometry (SAPHIR) has the worst performance. More precipitation events are detected with larger quantitative uncertainty in low-lying places than in highlands, in urban and water body areas than in other places, and more in coastal areas than in inland regions. Infrared (IR) estimate has worse performance than PMW, and the precipitation detectability of IR is more sensitive to the factors of elevation and the distance to the coast, as larger critical successful index (CSI) over lowlands and coastal areas. PMW morphing and the mixing of PMW and IR algorithms partly reverse the conservative feature of the precipitation detection of PMW and IR estimates, resulting in higher probability of detection (POD) and false alert ratio (FAR). Finally, monthly gauge calibration improves most of the statistical indicators and reduces the influence of elevation and surface type factor on these errors.

**Keywords:** IMERG; remote sensing; intermediate estimate; elevation; surface type

## 1. Introduction

Precipitation, one of the most fundamental hydrological processes in the water cycle, is the key factor in climate-related research, natural disaster forecasts, and hydrological applications [1]. High-quality precipitation data is the prerequisite for other "downstream" water processes. The traditional rainfall estimation relies on rain gauges, as a direct observation method, the observed precipitation record from which is widely considered as reliable. However, this gauge-based precipitation product suffers from problems, including sparse gauge coverage over oceans or remote regions, extensive labor and costs of maintenance and repair works, and inconsistent performance between gauges or in extreme weather [2].

The developments of remote sensing and digital computing technologies make the large-scale estimation of precipitation possible [3]. However, these indirect remote precipitation retrieval techniques also have their drawbacks. Despite the advantages, such as the high spatiotemporal resolution, because of the weak relationship between cloud top information and rain rates, the accuracy of precipitation data obtained from geosynchronous-Earth-orbit (GEO) infrared (IR) instruments is limited. On the contrary, the microwave sensors onboard low-Earth-orbit (LEO) platforms travel through clouds and therefore provide superior rainfall estimates, nevertheless inconsistent and limited sampling in time and space.

To enhance the performance and compensate for the shortcomings of different approaches, a few precipitation products merge various precipitation retrieval algorithms to generate multi-sensor products [1,4]. The Integrated Multi-satellitE Retrievals for the Global Precipitation Measurement mission (IMERG) algorithm is one of them, which combines information from gauges, microwave, and IR sensors to estimate precipitation over the entire Earth's surface [5,6]. Three runs of IMERG, namely the "Early", "Late", and "Final" runs, are provided based on latency and accuracy to accommodate different user needs [5,6].

Much research concerning the evaluation of IMERG performance has been conducted to specify the error types and sources [7–16]. Prakash et al. compared two Global Precipitation Measurement mission (GPM)-based products, IMERG and Global Satellite Mapping of Precipitation (GSMaP), with the TRMM Multi-satellite Precipitation Analysis (TMPA) product over India in 2016, and concluded that IMERG represents more realistic monsoon rainfall than the gauge-adjusted TMPA and GSMaP data, and all the three satellite-based products show larger errors over northeast India dominated by orographic effects [9]. Tan et al. (2016) leveraged ancillary variables in IMERG product to attribute the errors of IMERG to the individual microwave instruments, morphing, or IR retrieval algorithms, and reported more reliable estimates from passive microwave (PMW) than IR [10]. Wang et al. (2017) contrasted the hydrological utility of the IMERG near-real-time "Early" and "Late" runs with the post-real-time "Final" run exploiting the variable infiltration capacity (VIC) distributed hydrological model, and found higher accuracy of IMERG-Final than IMERG-Early and -Late [11]. In 2018, Wang et al. investigated the performance of three versions of IMERG (IMERG V03, V04, V05) and TMPA products, and found that IMERG has large areas of overestimation in the mountainous areas and underestimation in the coastal areas [12]. The precipitation estimate without gauge adjustment (IMERG_uncal) and the calibrated IMERG product (IMERG_cal) were compared with nine satellite and reanalysis products by Tang et al. [13]. Results show that IMERG product outperforms other datasets, except the Global Satellite Mapping of Precipitation (GSMaP) with daily gauge information.

However, most of the ground validation studies focus on the ultimate merged satellite-gauge precipitation estimate (IMERG_cal) and neglect the valuable intermediate estimates which directly guide the improvement of IMERG product. Moreover, little attention has been drawn to error identification of these intermediate estimates in different landscape conditions. This research aims to identify the error sources of IMERG version 6, final run, by evaluating the intermediate and ultimate estimates with the auxiliary variables, and to further illustrate the influences of regional topography and surface type on these errors for the better improvement of the next generation of merged precipitation products. The objectives of this work include (1) evaluation and comparison for the performance of the intermediate and ultimate estimates (HQprecipitation, IRprecipitation, precipitationUncal, precipitationCal) within IMERG version 6 product against the China Merged Perception Analysis (CMPA) dataset over the southeast coastal regions of China on three levels (a. the passive microwave estimates from different instruments; b. the merged PMW, morphed PMW, the mixture of morphing PMW and IR, and IR-only estimates; and c. the multi-satellite precipitation estimate without gauge adjustment and calibrated precipitation estimate), (2) investigation on the impact of orography on the errors of IMERG precipitation estimates, and (3) influences of surface types (urban, water bodies, and non-urban, non-water regions) and the distance to the coast on the performance of IMERG products.

This study aims to provide first-hand and detailed evaluations of IMERG product to researchers, algorithm developers, and end users when considering the limitations of the current product and the improvements of the next generation. This paper is structured as follows: Section 2 describes the study area, datasets, and research methods. The research results corresponding to three research objectives are indicated in Section 3. Finally, the relevant discussion and main construction are explained in Sections 4 and 5.

## 2. Materials and Methods

### 2.1. Study Region

The southeast coastal areas of China were selected as the study region, extending from 109°E to 122°E in longitude and from 18°N to 33°N in latitude, with pockets of lower coastal plains and interior mountainous areas, as shown in Figure 1. The elevation in study areas varies between 0 and 1684 m, and the climate is dominated by the subtropical monsoon. The annual precipitation is about 2000 mm, while influenced by summer monsoon, most of the precipitation happens in the wet season between May and September. During the summer monsoon period, the high elevation mountains block the southeast wind which brings moist air from the warm sea to the inland place and facilitates the orographic convective precipitation events at the windward slopes. Therefore, the annual precipitation at southeast windward can reach 2400 mm, significantly larger than the leeward sides.

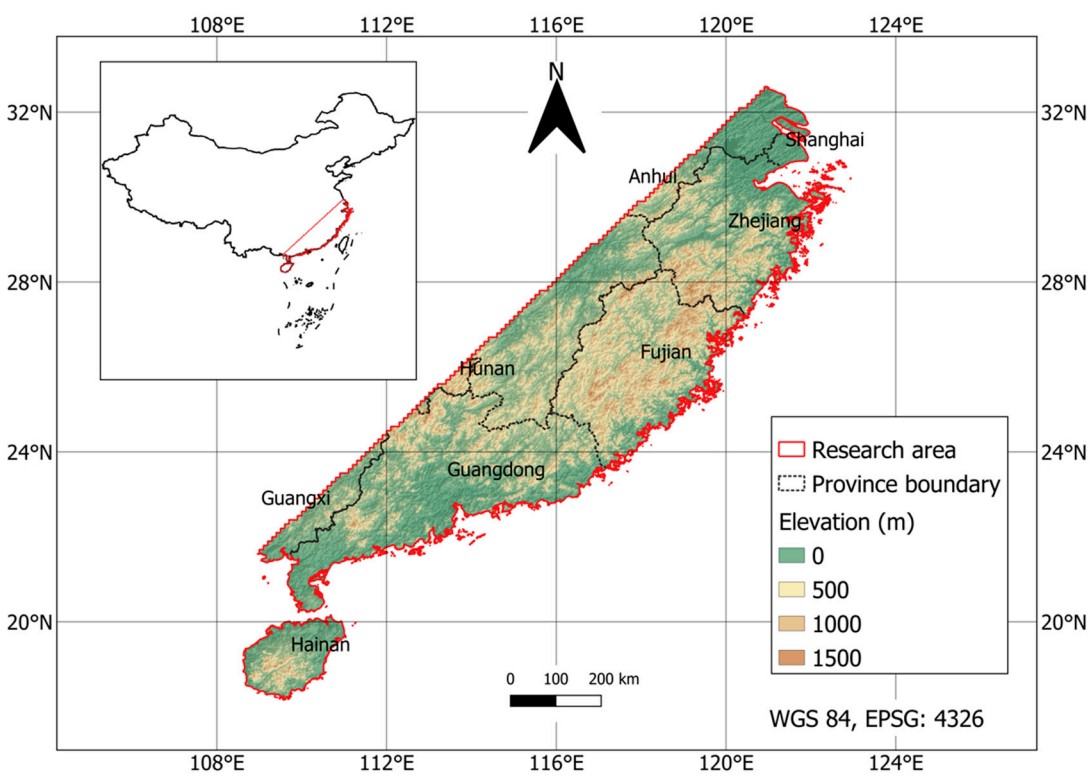

**Figure 1.** Orographic information of the study region.

### 2.2. Dataset

#### 2.2.1. IMERG

The Integrated Multi-satellitE Retrievals for GPM (IMERG) is level-3 gridded precipitation data, which is intended to provide the multi-satellite precipitation product over the entire globe, combining the estimates collected from "all" microwave and infrared sensors, precipitation gauge analyses, together with the climate reanalysis products. Served by the TRMM and GPM satellite constellations,

the latest version 06 of IMERG, released in 2019, is available at 0.1° spatial resolution and half-hourly time intervals. Three modes of IMERG products are provided for different utilities with distinct latencies and accuracies as early, late and final runs. In this study, the post-real-time final run with high accuracy, produced after about a 3.5-month observation time and intended for scientific research uses, is concerned [5,6].

The main contributions of the precipitation estimate in IMERG V06 product are retrieved from the GPM satellite PMW instruments, with the 2017 version of the Goddard Profiling Algorithm (GPROF2017) for most PMW sensors and Precipitation Retrieval and Profiling Scheme (PRPS) [17] for Sounder for Atmospheric Profiling of Humidity in the Intertropics by Radiometry (SAPHIR). After being intercalibrated by the GPM Combined Radar Radiometer Analysis (CORRA) product and climate model, the high-quality PMW estimate is combined into half-hourly and 0.1° fields. This is provided to the -Precipitation Estimation from Remotely Sensed Information using Artificial Neural Networks–Cloud Classification System (PERSIANN-CCS) [18] infrared (IR) re-calibration procedure with the zenith-angle-corrected and intercalibrated merged geo-IR fields Climate Prediction Center (CPC) assembled. Then, PMW high quality and PERSIANN-CCS IR estimates are sent to the CPC Morphing-Kalman Filter (CMORPH-KF) [19] quasi-Lagrangian time interpolation scheme, with either the Goddard Earth Observing System model-Forward Processing (GEOS-FP) for the early run or Modern Era Retrospective Reanalysis 2 (MERRA-2) for late and final runs. At last, the multi-satellite estimate was calibrated with the Global Precipitation Climatology Centre (GPCC) monthly precipitation gauge analysis for bias adjustments as the "Final" satellite-gauge product [5,6].

### 2.2.2. CMPA

China Merge Precipitation Analysis hourly V1.0 product (CMPA) [20], a gauge-satellite merged gridded precipitation estimate developed by the National Meteorological Information Center of the China Metrological Administration (CMA), was used as reference data for the performance evaluation of IMERG product. The hourly rain gauge data at more than 30,000 automatic weather stations over mainland China and the half-hourly, 8 km Climate Precipitation Center Morphing (CMORPH) product [21] were exploited to merge daily, 0.25° and hourly, 0.1° gauge-satellite CMPA precipitation data, using the probability density function-optimal interpolation (PDF-OI) scheme.

The high spatial and temporal resolution of CMPA allowed us to assess the IMEGR product at an hourly time scale and 0.1° spatial scale. In addition, high-density gauge-based precipitation observation in southeast China guarantee the accuracy and reliability of CMPA product [20,22]. To compare with the resolution of CMPA product, the half-hourly IMERG data was accumulated hourly in this study.

### 2.2.3. Topography and Surface Type Data

The topography and surface type information was retrieved from the National Aeronautics and Space Administration (NASA) Earth Observations (https://neo.sci.gsfc.nasa.gov/). The Digital Elevation Model (DEM) was produced based on three data sources, including NASA's Space Shuttle, Canada's Radarsat satellite, and topographic maps made by the United States Geological Survey. The surface type was derived from the Moderate Resolution Imaging Spectroradiometer (MODIS) land-cover products, which constructs 17 categories of land cover and was processed into 7 categories in this study. The spatial resolution of both the DEM and surface type is 0.1 degree, the same as the IMERG product. The maps regarding surface type and the distance from the sea are shown in Figure 2.

### 2.3. Methodology

In this study, three intermediate and one ultimate precipitation products within IMERG products were processed into 12 categories of precipitation estimates for three levels of statistical evaluation. The specific assessment procedures were organized into three parts corresponding to three objectives. First, a general assessment was conducted by qualitatively and quantitatively comparing 12 categorical IMERG estimates with the CMPA product. Then, five elevation bands were exploited to divide the

IMERG and CMPA data to glance at the topographic influences on the errors of 12 IMERG estimates. Finally, the effects of surface type factors, including urban, water bodies, or other areas, and the distance to the coastline on the error components of IMERG products were analyzed.

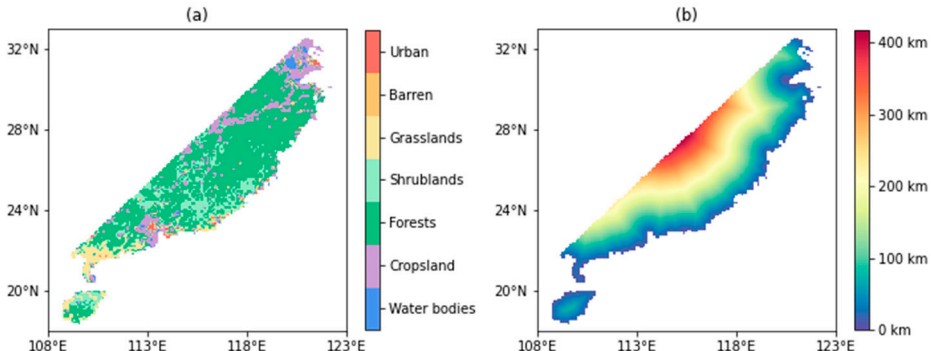

**Figure 2.** Surface types (**a**) and the distance to the sea (**b**) in study region.

### 2.3.1. Variables Processing

Six variables, as three intermediate and one ultimate estimates, as well as two auxiliary variables, including HQprecipitation, IRprecipitation, precipitationUncal, precipitationCal, HQprecipSource, and IRkalmanFilterWeight, were exploited to thoroughly evaluate and identify the error source of IMERG product at three levels. The variable names and their definitions can be found in Table 1.

**Table 1.** List of used variable names, their definition, and units in the Integrated Multi-satellitE Retrievals for the Global Precipitation Measurement mission (IMERG) products.

| Variable Name | Definition | Units |
|---|---|---|
| HQprecipitation [1] | Merged microwave-only precipitation estimate | mm/h |
| IRprecipitation | IR-only precipitation estimate | mm/h |
| precipitationUncal | Multi-satellite precipitation estimate | mm/h |
| precipitationCal | Multi-satellite precipitation estimate with gauge calibration | mm/h |
| HQprecipSource | Microwave satellite source identifier | - |
| IRkalmanFilterWeight | Weights of IR-only precipitation relative to the morphed merged microwave-only precipitation | % |

[1] HQprecipitation has significant spatial and temporal gaps, as merged passive microwave (PWM) observation data.

For the first level of IMERG product evaluation, concerning the different performance of six PMW instruments, the field of HQprecipitation needs to be divided according to different PMW sources identified in HQprecipSource variable. The index values for HQprecipSource, the name of the corresponding PWM instrument, and the data amounts and weights contributed by each PWM sensors involved in this research are displayed in Table 2.

**Table 2.** List of the passive microwave (PMW) instruments and the data amount in this research.

| HQprecipsource | Sensor Type | PMW Instrument | Amount of Data | Weight of Data |
|---|---|---|---|---|
| 3 | Imager | Advanced Microwave Scanning Radiometer Version 2 (AMSR-2) | 542,790 | 9.5% |
| 5 | Imager | Special Sensor Microwave Imager/Sounder (SSMIS) | 1,612,327 | 28% |
| 7 | Sounder | Microwave Humidity Sounder (MHS) | 2,139,939 | 37% |
| 9 | Imager | GPM Microwave Imager (GMI) | 325,118 | 5.7% |
| 11 | Sounder | Advanced Temperature and Moisture Sounder (ATMS) | 683,712 | 12% |
| 20 | Sounder | Sounder for Atmospheric Profiling of Humidity in the Intertropics by Radiometry (SAPHIR) | 422,434 | 7.4% |

For the second level of assessment, five variables were exploited to identify four types of estimates which include the merged PMW-only, morphed PMW, the mixture of morphed PMW and IR, and IR-only estimates. Firstly, the HQprecipitation and IRprecipitation provided the merged PMW-only and IR-only estimates, respectively. When there was no direct PMW observation (HQpreciSource = 0), IRkalmanFilterWeight (from 0 to 100), indicating the weight percentage of IR-only data to the morphing merged PMW-only estimation, was used to divide the precipitationUncal: If the IRkalmanFilterWeight was equal to 0, the precipitationUncal was classified as morphing PMW; If the IRkalmanFilterWeight was larger than 0 and smaller than 100, the precipitationUncal was categorized into the mixture of morphing PMW and IR group. It should be noticed that the IRkalmanFilterWeight was always smaller than 100 in this research area and period, which means the IR-only estimation (IRprecipitation) was not used directly in the multi-satellite estimation without gauge calibration layer (precipitationUncal). The amounts (and weights) of the data sample collected from the three components, merged PMW-only, morphed PMW, and the mixture of morphed PMW and IR, in the precipitationUncal estimate were 5,726,320 (29%), 7,502,932 (38%), 6,448,996 (33%).

Finally, the performances of uncalibrated multi-satellite (precipitationUncal) and monthly gauge calculated satellite-gauge (precipitationCal) estimations were compared in the last level of evaluation. Because of the spatiotemporal discontinuity of PWM data and the division of the precipitationUncal field for the second layer of assessment, the spatial distributions of sampling frequency for nine categories, containing six PMW sensors, merged PMW-only, morphing PMW, and the mixture of morphing PMW and IR, are inconsistent as shown in Figure 3.

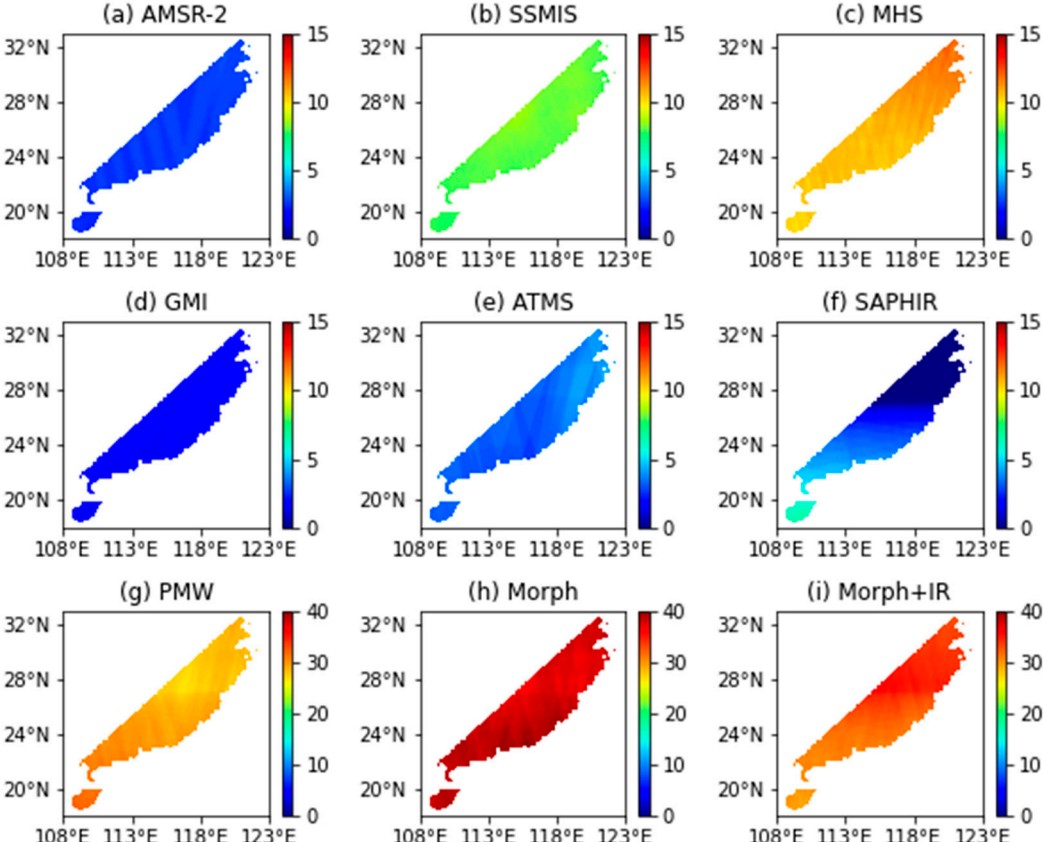

**Figure 3.** Spatial distributions of sampling frequency for nine categories in percentage, containing (**a–f**) six PMW sensors, (**g**) merged PMW, (**h**) morphed PMW, and (**i**) the mixture of morphing PMW and Infrared (IR).

2.3.2. Statistical Evaluation Metrics

The CMPA gridded data were clipped and used as the ground truth to assess the 12 IMERG estimates with different spatiotemporal distributions. Because of the spatial and temporal discontinuity of most of the 12 estimates, this research focused on the error comparison among 12 IMERG precipitation datasets instead of detecting the continuously spatial or temporal error patterns of each product.

The evaluation of IMERG product was conducted in two procedures focusing on qualitative precipitation detection and quantitative rain rate estimation, respectively. Since the extreme light rain rate of both IMERG and CMPA products bear great uncertainty, a commonly used threshold of 0.1 mm/h was exploited to divide the rain and not-rain events [10,23].

The assessment for precipitation detectability was carried out with the contingency tables in regards to the number of rain and no rain judgments by IMERG and CMPA products respectively, as well as three categorical scores including probability of detection (POD), false alert ratio (FAR), critical successful index (CSI). The fractions of the IMERG hit, miss, false alarm, and true negative detections in contingency tables aim to provide a basic and direct performance comparison among 12 test estimates. POD and FAR focus on type II and I errors within the IMERG product, while CSI is a composite score reflecting both of the two types of errors.

After the qualitative evaluation of precipitation detection, the hit events in IMERG product were quantitatively assessed. In this process, four continuous indices were used, such as mean error (ME), root-mean-square error (RMSE), BIAS, and correlation coefficient (CC). ME quantifies the mean difference between the IMERG and CMPA values, RMSE and BIAS indicate the random and systematic components of errors, and the CC denotes the correlation between the IMERG and CMPA products [10,11,15,24,25]. These statistical metrics were calculated according to the formulas in Table 3.

**Table 3.** List of the statistical metrics for Integrated Multi-satellitE Retrievals for the Global Precipitation Measurement mission IMERG (IMERG) evaluation.

| Type | Metric | Formula | Unit | Optimal Value |
|------|--------|---------|------|---------------|
| Categorical index | Probability of Detection (POD) | $\text{POD} = \frac{H}{H+M}$ | - | 1 |
| | False Alert Ratio (FAR) | $\text{FAR} = \frac{F}{H+F}$ | - | 0 |
| | Critical Successful Index (CSI) | $\text{CSI} = \frac{H}{H+M+F}$ | - | 1 |
| Continuous index | Mean Error (ME) | $\text{ME} = \frac{\sum_{i=1}^{n}(y_i - x_i)}{n}$ | mm/h | 0 |
| | Root-Mean-Square Error (RMSE) | $\text{RMSE} = \sqrt{\frac{1}{n}\sum_{i=1}^{n}(y_i - x_i)^2}$ | mm/h | 0 |
| | BIAS | $\text{BIAS} = \frac{\sum_{i=1}^{n}(y_i - x_i)}{\sum_{i=1}^{n} x_i}$ | - | 0 |
| | Correlation Coefficient (CC) | $\text{CC} = \frac{\sum_{i=1}^{n}(x_i - \bar{x})(y_i - \bar{y})}{\sqrt{\sum_{i=1}^{n}(x_i - \bar{x})^2(y_i - \bar{y})^2}}$ | - | 1 |

*H*, *M*, and *F* indicate the numbers of the hit, miss, and false alarm events; $x_i$ and $y_i$ demonstrate the reference and predict rain rates in CMPA and IMERG products respectively; and *n* denotes the total number of the hit events.

## 3. Results

### 3.1. General Assessment

Figure 4 shows the evaluation results on precipitation detection for 12 IMERG estimates (including six individual PMW sensors, merged PMW, morphed PMW, the mixture of morphed PMW and IR, IR-only, and the multi-satellite estimates without and with gauge adjustment). The hit and false alarm percentages for 12 estimates vary significantly, with the ranges of hits from 4.8% to 8.4% and of false alarms from 4.7 to 10.8%, and both of the maximum and minimum fractions are contributed by the

individual PWM instruments. As for the misses, the performances of 12 estimations are more stable varying between 4.8% and 7.7%.

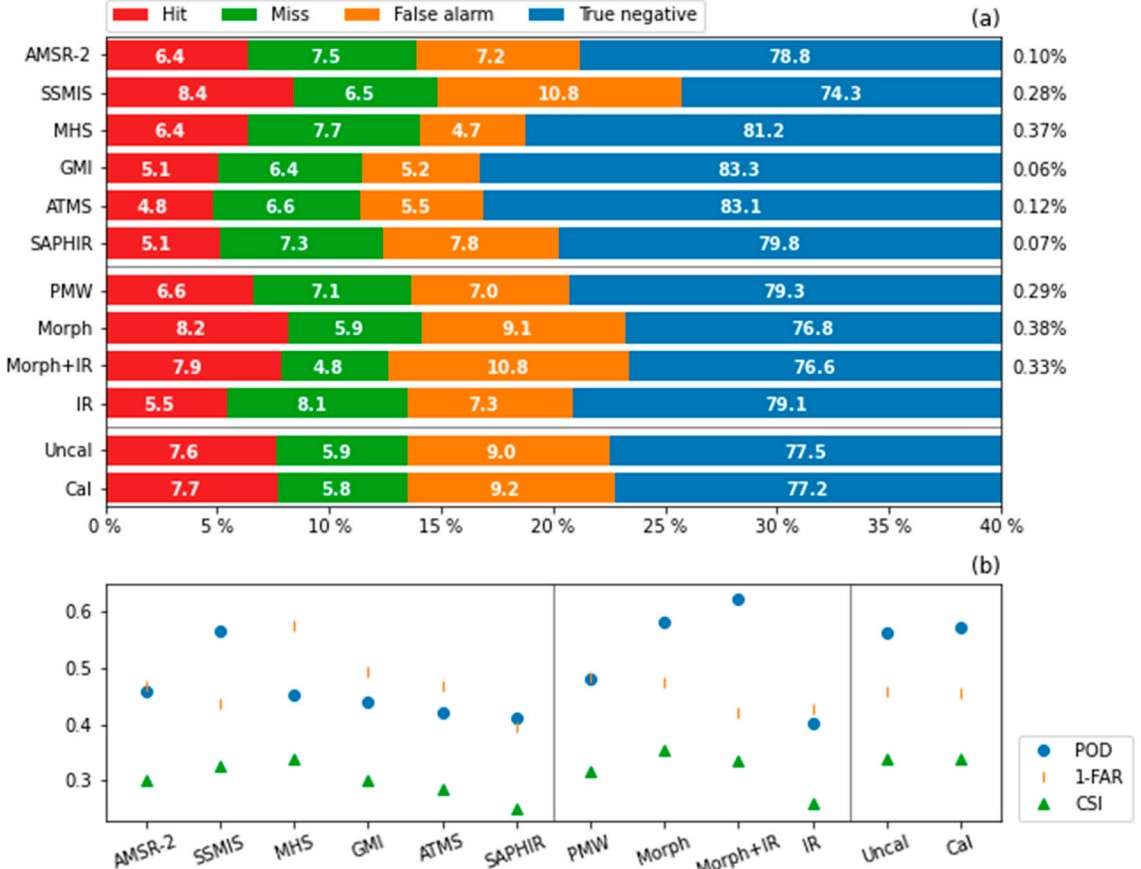

**Figure 4.** Detectability of precipitation events for 12 IMERG estimates verified against the China Merge Precipitation Analysis hourly V1.0 product (CMPA) product (**a**) Hit, miss, false alarm, and true negative fractions in percentage. The right column of proportion indicates the sampling weights in the corresponding "PMW" or "Uncal" data. (**b**) Three categorical indices, probability of detection (POD), false alert ratio (FAR), and critical successful index (CSI), for 12 IMERG estimates.

The characteristics of different PMW sensors were studied by comparing six individual PMW estimates. ATMS was prone to neglect precipitation events with the least hit proportion (4.8%) and a relatively low false alarm ratio (5.5%). GMI suffered from a similar drawback, but had better performance with a larger fraction of hits (5.1%) and fewer false alarms (5.2%). By contrast, the precipitation detectability of SSMIS far surpassed other PWM sensors with the largest hit percentage (8.4%), despite the high false alarm ratio (10.8%), which may be suitable for storm warning. Although suffering from the type I error with a high missed fraction (7.7%), MHS had superior comprehensive performance among six PWM sensors because of high hits (6.4%) and lowest false alarms fractions (4.7%). SAPHIR performed worst compared with the other five PWM sensors with high proportions of misses (7.3%) and false alarms (7.8%), and fewer hits (5.1%), notwithstanding the use of the separate Precipitation Retrieval and Profiling Scheme (PRPS) [17]. This may be because the insufficient data samples in the study area (only taking 0.07% of all PMW observed data) leads to unstable prediction performance. AMSR-2 had a moderate performance with a satisfying hit fraction (6.4%), but both of the two kinds of errors are significant (7.5% for misses and 7.2% for false alarms).

For the second level of evaluation, the data from six PWM sensors were merged and compared with morphed PMW, the mixture of morphed PMW and IR, and IR-only estimates. Unsurprisingly, the IR estimate performed worst in this group, with two kinds of errors in occuring frequently, as 8.5%

misses and 7.3% false alarms. The merged PMW had better performance but was over-conservative on precipitation identification with small proportions of hits (6.6%) and false alarms (7.0%), and a large miss fraction (7.1%). After morphing, the conservative tendency of PMW observations was corrected partly with more hits (8.2%) and fewer misses (5.9%), at the price of more false alarms (9.1%). Then, the combing of morphed PMW and IR algorithms further reduced the miss percentage to 4.8% but enlarged the false alarms to 10.8%.

In general, the differences were small between the multi-satellite estimates with and without the gauge calibration for the last level of assessment, but the conservative feature was slightly reversed again, with larger hit and false alarms fractions after the gauge calibration.

After the assessment concerning the detectability of IMERG precipitation product, the hit events in IMERG product were quantitatively assessed. Figure 5 shows the distribution comparisons of the hit pairs of rain rate data for 12 IMERG estimates and corresponding CMPA data, and Table 4 indicates their statistical information. It can be found that six conditional mean rain rates of IMERG PMW estimates fall between 2.29 and 3.35 mm/h, while the CMPA vary slightly from 2.45 to 3.18 mm/h, with the mean error ranging from −0.48 mm/h for SSMIS to 0.84 mm/h for ATMS. In addition, most of the PMW sensors tended to overestimate light (<1 mm/h) and extremely large rain rates (>50 mm/h).

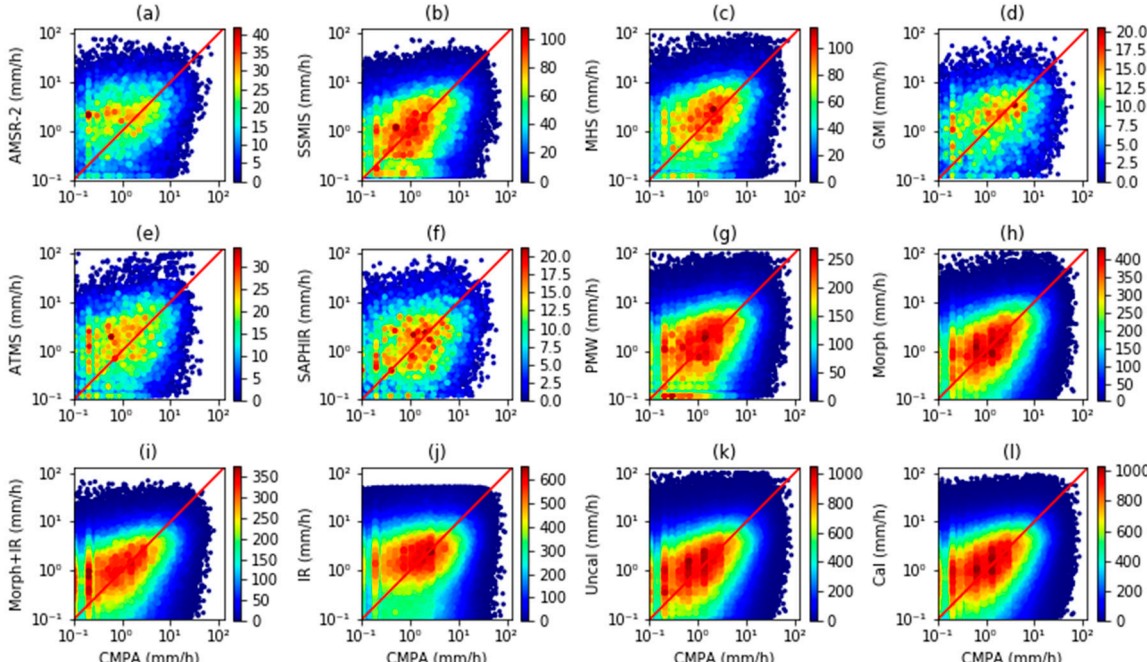

**Figure 5.** Distribution density of hit pairs of data for 12 IMERG estimates, including (**a**–**f**) six individual PMW, (**g**) merged PMW, (**h**) morphed PMW, (**i**) the mixture of morphed PMW and IR, (**j**) IR-only, (**k**) without and (**l**) with gauge adjustment estimates, and corresponding CMPA data in logarithmic scale. The statistics of them are indicated in Table 4.

Among six test PMW sensors, MHS, with sufficient hit data, had the best performance, with most of the scatter points gathering near the diagonal, which obtained the highest CC score (0.25) and relatively low systematic (with the bias as 3.9%) and random errors (with the RMSE as 5.8 mm/h) but tended to report extremely large rain rates. SSMIS, as the instrument detecting most hit faction of precipitation events, had a strong correlation with CMPA data (with CC as 0.25) and minimal random error (with the RMSE as 4.8 mm/h), yet it suffered from a general underestimation, with the bias as −17%. Both AMSR-2 and ATMS produced considerable overestimations, with the highest ME (0.81 mm/h and 0.84 mm/h) and bias (32% and 34%), while their CC and RMSE of are at medium level. Similar to AMSR-2 and ATMS, GMI significantly overestimated light rain events as shown in Figure 5d, but the medium rain rates around 10 mm/h were underestimated, both of which resulted in

the most neutral bias as −2.5%. In addition, the RMSE score gives a glance at the random errors of GMI, which was 6.3 mm/h, as the highest RMSE amongst six PWM sensors. For SAPHIR, the random error was also significant, as 6.2 mm/h for RMSE, and the correlation with the CMPA product was worst, as 0.14 for CC, yet the systematic error was not severe compared with the other five PMW instruments, as 8.4% for bias.

**Table 4.** Statistical comparison of the hit data pairs for 12 categories of IMERG estimates and corresponding CMPA data.

| | Amount of Hit Samples | Conditional Mean Rain Rate for CMPA (mm/h) | Conditional Mean Rain Rate for IMERG (mm/h) | Conditional ME (mm/h) | Conditional RMSE (mm/h) | Conditional BIAS (%) | Conditional CC |
|---|---|---|---|---|---|---|---|
| AMSR-2 | 34,714 | 2.54 | 3.35 | 0.81 | 5.7 | 32 | 0.18 |
| SSMIS | 135,627 | 2.77 | 2.29 | −0.48 | 4.8 | −17 | 0.25 |
| MHS | 136,174 | 3.01 | 3.13 | 0.12 | 5.8 | 3.9 | 0.25 |
| GMI | 16,448 | 3.18 | 3.10 | −0.08 | 6.3 | −2.5 | 0.15 |
| ATMS | 32,950 | 2.45 | 3.29 | 0.84 | 6.0 | 34 | 0.23 |
| SAPHIR | 21,529 | 2.83 | 3.06 | 0.24 | 6.2 | 8.4 | 0.14 |
| PMW | 377,442 | 2.83 | 2.86 | 0.03 | 5.5 | 1.0 | 0.22 |
| Morph | 617,039 | 2.74 | 2.45 | −0.29 | 4.8 | −11 | 0.27 |
| Morph +IR | 507,484 | 2.58 | 2.10 | −0.48 | 4.3 | −19 | 0.28 |
| IR | 2,629,350 | 3.08 | 2.9 | −0.18 | 5.7 | −5.7 | 0.16 |
| Uncal | 2,661,065 | 2.71 | 2.44 | −0.27 | 4.8 | −10 | 0.25 |
| Cal | 2,661,065 | 2.69 | 2.49 | −0.20 | 4.7 | −7.6 | 0.25 |

For the second level of evaluation among the merged and morphed PMW, the mixture of morphed PMW and IR, and IR-only estimates, although directly retrieved from satellites, merged PMW and IR-only estimates, provided ostensibly reliable precipitation data with less systematic error as 1.0% for PMW and −5.7% for IR as shown in Table 4, their distinct features need further examination. PMW tended to overestimate light (<1 mm/h) and extreme large rain rates (>50 mm/h), but on the other hand, it underestimated the medium rain rates (around 20 mm/h), both of which led to a seemingly low bias as a whole. IR estimates were all capped at 54 mm/h abnormally, however, which number almost doubles to 104 mm/h for the corresponding CMPA category, and the light rain rates (<1 mm/h) were overestimated appreciably, which balances the absence of the large rain events. Because of quite high conditional mean rain rates, both PMW and IR had large random errors (with the RMSEs as 5.5 mm/h for PMW and 5.7 mm/h for IR), as well as weak correlation with CMPA (with the CCs as 0.22 for PMW and 0.16 for IR). However, the morphing algorithm and the mixture of IR gradually improved the consistency, with the CC increasing from 0.22 (for merged PMW) to 0.27 (for morphed PMW) and finally 0.28 (for the mixture of morphed PMW and IR). Moreover, with the shrinkage of conditional mean rain rate, the random error diminished with the RMSE falling from 5.5 mm/h (for merged PMW) to 4.8 mm/h (for morphed PMW) and 4.3 mm/h (for the mixture of morphed PMW and IR). As for the systematic error, the overestimation for large rain rates of PMW estimate was alleviated partly after morphing, which caused an overall negative bias for morphed PMW estimates as −11%. Then, the mixture of IR entirely offset the overestimation problem on extreme large rains for PMW estimates, however, this further intensified the negative bias to −19%.

Finally, for the last two multi-satellite and satellite-gauge estimates, the properties of overestimation for light and extremely large rain rains of IR and PMW estimates were retained, and the overestimation on extreme large rains was mitigated after gauge calibration, as shown in Figure 5k,l. Except for the CC remaining at 0.25, all the statistics indices were improved.

### 3.2. Topographic Influence

The precipitation detectability of 12 IMERG estimates was studied according to different elevation bands, as shown in Figure 6. In general, more precipitation events were detected in lower places than highlands, as most of the POD and FAR indices decrease with the elevation rising.

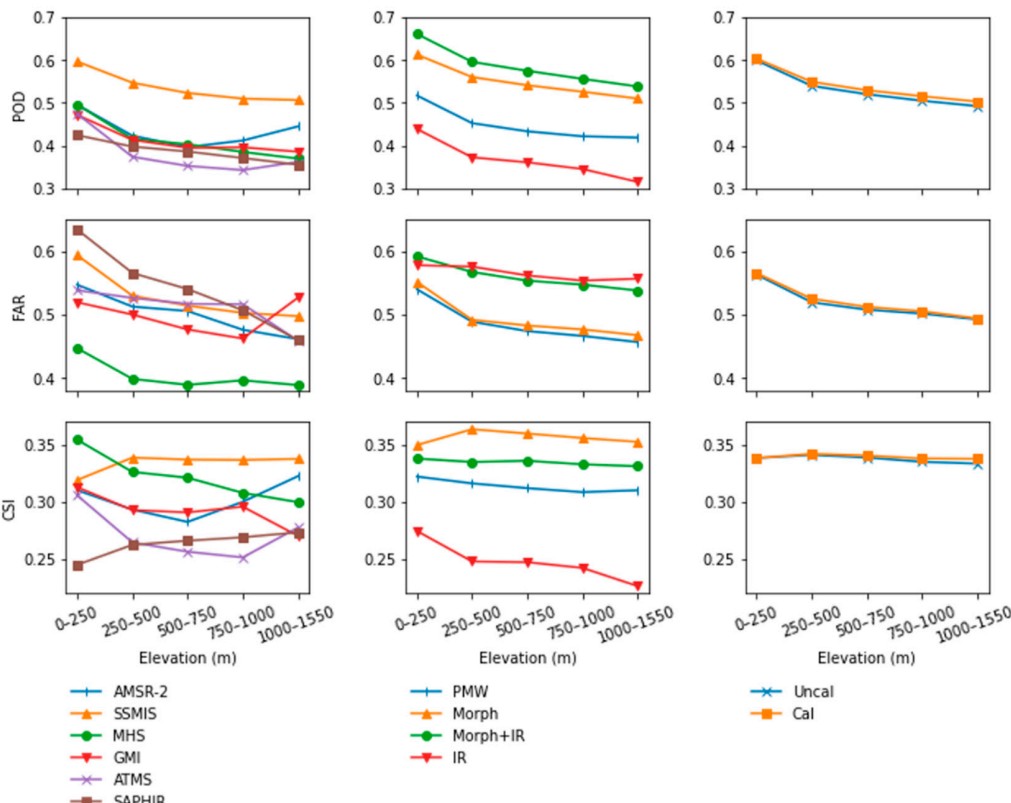

**Figure 6.** Three categorical indices, higher probability of detection (POD), false alert ratio (FAR), and critical successful index (CSI), for 12 IMERG estimates according to different elevation bands. The left column indicates six individual PMW estimates, and the middle column demonstrates four multi-satellite estimates for the second level of evaluation, among the merged PWM, morphed PWM, the mixture of morphed PMW and IR, and IR-only estimates, and finally, the right column shows the multi-satellite estimates with and without gauge calibration.

For the first level of evaluation, most of the PMW sensors had larger POD and FAR scores in lowlands, excepting a few irregular indexes for the highest altitude group without a sufficient data sample. Beyond that, the POD score for SSMIS and FAR score for MHS far surpassed the other five categories at all elevation levels, which gave them extraordinary overall performance with high CSIs. In summary of the CSI score, the performance of SSMIS was relatively stable for all elevation bands. In lowlands (0–250 m), except for SAPHIR, all the PMW sensors had satisfactory performances with the CSI larger than 0.3, while in highlands (>1000 m) SSMIS, AMSR-2, as well as MHS were more reliable than others.

For the second level of evaluation concerning the merged and morphed PMW, the mixture of morphed PMW and IR, and IR-only estimates, the POD scores for the lowest group (0–250 m) were about 0.1 higher than the highest group (>1000 m) for all the four categories of IMERG estimates. As for FAR, the merged and morphed PMW estimates were more sensitive to the elevation factor than the IR estimate, as the FAR index decreased by 0.08 with elevation rising, which range was 0.02 for IR and 0.05 for the mixture of morphed PMW and IR. For the overall performance regarding the CSI index, IR performed worse, with lower CSI in highlands than lowlands, however, which was relatively stable for the other three IMERG estimates.

By the last level of comparison between two multi-satellite estimates without (Uncal) and with (Cal) gauge calibration, it was found that the monthly gauge calibration slightly increased the POD in high lands and therefore resulted in marginally higher CSI.

For the hit estimates, the quantitative evaluation of the elevation influences on 12 IMERG estimates was conducted using four continuous indices, RMSE, ME, BIAS, and CC, as shown in Figure 7. It is obvious that lower areas had larger conditional mean rain rates, and therefore suffered from larger random errors than the higher places on the whole.

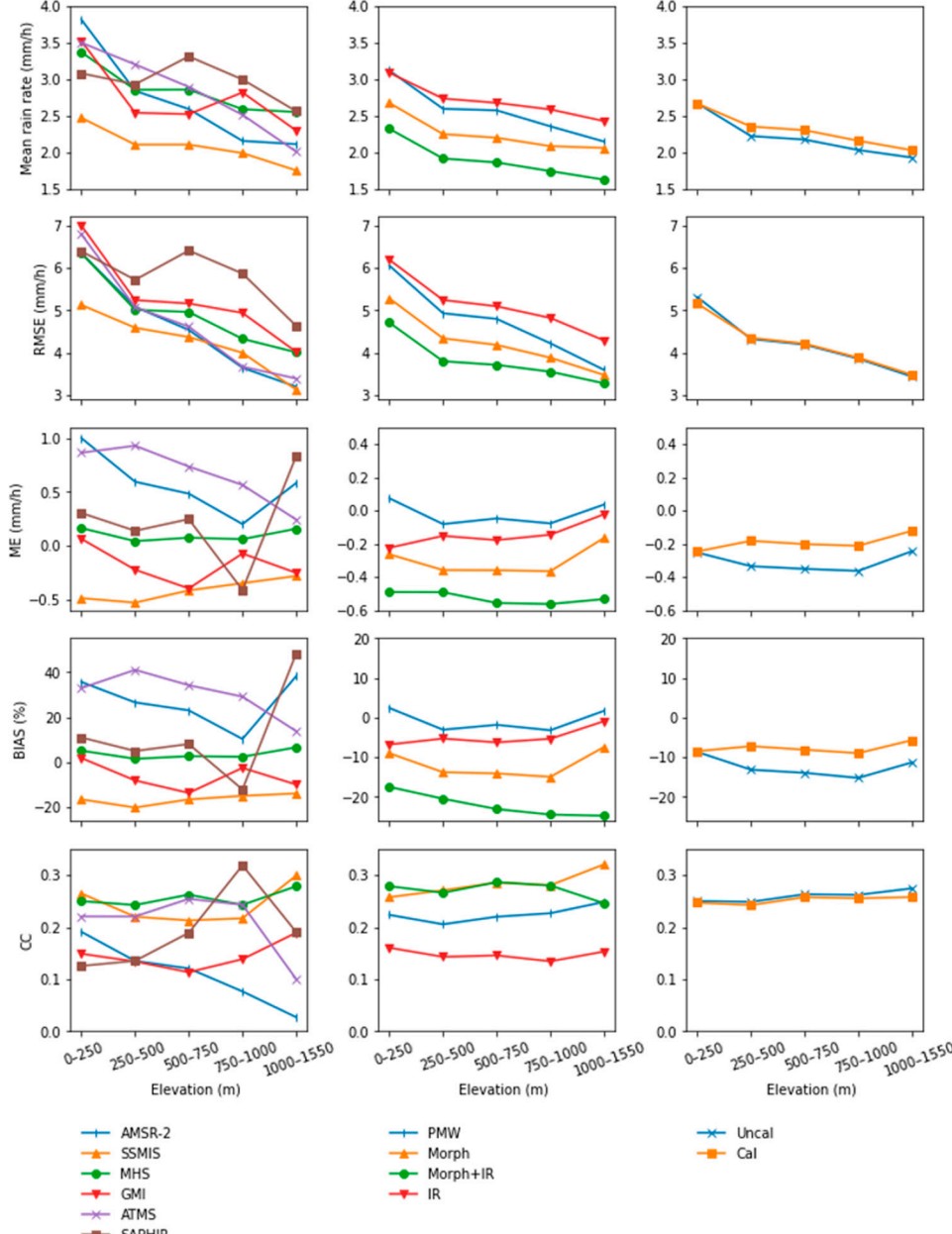

**Figure 7.** Mean value and four continuous indices, Root-Mean-Square Error (RMSE), Mean Error (ME), BIAS, and Correlation Coefficient (CC), of the hit data of 12 IMERG estimates according to different elevation bands. The left, middle, and right columns indicate the IMERG estimates for the first, second, and third levels of evaluations. It is noted that the scales of ME and BIAS in the left figures are different from their values in the middle and right figures, since the different systematic performances of the evaluated products.

In six PMW instruments, AMSR-2 was influenced most by the elevation factor, with the largest difference in the conditional mean rain rate as 1.7 mm/h, whereas SSMIS had the least difference, at 0.7 mm/h. The random error had a strong positive correlation with the mean rain rate. As for the systematic error, most of the biases for six PMW estimates from five elevation groups were between 40% and −20%. MHS and SSMIS had relatively stable biases, around 3.6% and −16.4% for all elevation groups. The bias of GMI varied between 1.8% and −13.7%, while both AMSR-2 and ATMS had considerable overestimation for all elevation groups especially in lower places. SAPHIR had positive biases at altitudes below 750 m, while it significantly underestimated from 750 to 1000 m. Several anomalous bias values for the largest altitude group may be due to the small amount of data from individual PMW sensors. Regarding the correlation with CMPA, MHS and SSMIS still had stable and satisfactory performances for five altitude groups, compared with other PMW sensors. Moreover, SSMIS and GMI displayed higher CC scores in the lowest and highest elevation bands than in the medium band, while the CC index for AMSR-2 declined significantly, from 0.19 to 0.03, with the rise in elevation. The CC of SAPHIR fluctuated most with the varying altitude, with larger CC in high places.

Among the four estimates for the second level of evaluation, the elevation factor had larger influences on the merged PMW estimate, with the largest differences of conditional mean rain rate and RMSE between the highest and lowest bands at 0.97 and 2.46 mm/h, while these numbers for the other three categories were lower than 0.70 and 1.90 mm/h. As for the systematic error, IR had more obvious underestimation in lower regions, while the mixture of IR and morphed PMW performed conversely. In addition, the merged and morphed PMW had lower biases for the medium elevation groups (250–1000 m) than for the lowest and highest. For the correlation with CMPA, the merged and morphed PMW again shared a similar characteristic: that of greater CC in higher places, which feature was further propagated to the final estimate. However, the CC values of IR and the mixture of morphed PMW and IR were not effected obviously by the elevation factor.

Finally, the monthly gauge calibration significantly corrected the underestimate in the places higher than 250 m with the increase of negative biases by 6%, and mitigates the elevation influences on both the systematic and random errors with less bias and RMSE differences between the highest and lowest bands.

### 3.3. Influence of Surface Type

#### 3.3.1. Urban and Water Body

The detectability of 12 types of IMERG estimates in urban, water, and other areas is analyzed in Figure 8. In general, more precipitation events were recognized in urban and water areas with larger POD and FAR scores than other places, similar to the study of Tian and Peters-Lidard [26] over bodies of water. In addition, non-urban, non-water areas had a better comprehensive performance with a higher CSI score than the urban and water bodies.

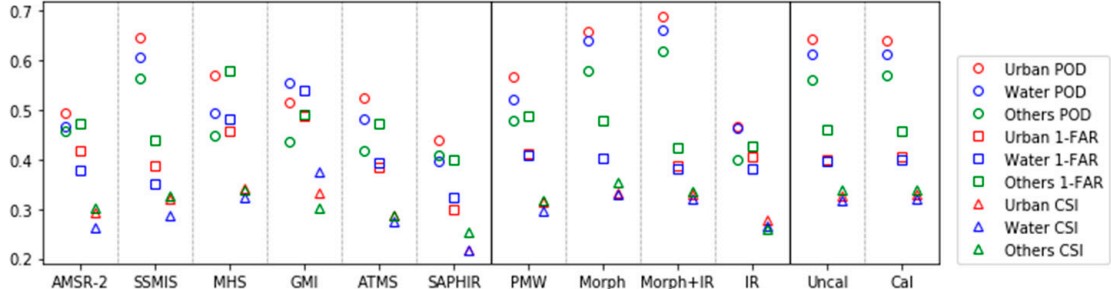

**Figure 8.** Three categorical indices, POD, FAR, and CSI, for 12 IMERG estimates in urban, water, and other areas.

For most of the six individual PMW sensors in the first level of evaluation, urban areas benefited from the highest POD score, followed by bodies of water, and finally other areas. The POD indexes for four instruments, SSMIS, MHS, GMI, and ATMS, clearly changed, according to surface types with ranges larger than 0.1, while AMSR-2 and SAPHIR were influenced less. As for FAR, the non-urban, non-water places showed more desirable performance, with lower false alarm ratios than for the urban and water areas for all the PMW sensors, except for GMI. Considering both two types of error regarding false positives and negatives, the CSI indexes in water areas, for all PMW instruments except for GMI, were lower than in other places, but GMI had an especially extraordinary performance in water body areas, among the three surface type categories, with the most hit and least false alarm fractions.

Moreover, four types of estimates in the second level of evaluation had higher PODs and FARs in urban and water areas than others. Comparing the urban and water areas, the merged PMW had 0.05 higher POD in urban districts than water body areas, while the difference was negligible for IR. With the morphing and PMW-IR mixing algorithms, more precipitation events were detected, resulting in the increase of PODs for all surface types at the expense of larger FAR, especially in non-urban, non-water areas. For the CSI indicator, there were a few differences among the three categories of surface type, but the IR-only estimate unexpectedly had higher CSI in urban and water areas than others, which is contrary to other PMW-involved estimates.

For the last level of evaluation concerning the uncalibrated and calibrated multi-satellite estimates (Cal and Uncal), the monthly gauge calibration reduced the differences among the three categories of surface type. The POD score in non-urban, non-water districts increased, at the cost of lower PODs in urban and water areas, and the FAR scores in urban and water areas decreased, partly counteracting a larger FAR in other places. In general, the CSI scores for all surface types increased slightly.

Next, the hit events were further quantitatively evaluated with four continuous indexes according to different surface types as shown in Figure 9. Among six PMW estimates, urban areas bore the largest uncertainty in the precipitation estimates, with the widest range of conditional mean rain rates, followed by water bodies, while other places had a stable performance with sufficient data samples. Four PMW estimates, GMI, AMSR-2, MHS, and SSMIS, observed more intense precipitation in urban and water areas than other places, while, abnormally, the non-urban, non-water districts became the rainiest places for SAPHIR. Generally speaking, the RMSE index had a strong positive correlation with the mean rain rate, but compared with imagers, three sounders had more consistent performances on random errors for different surface types. As for the systematic error, two sounders, MHS and SAPHIR, had better and stable performances with low biases for different surface types. Except for SSMIS and SAPHIR, most PMW sensors had higher CCs in urban areas, followed by non-urban and non-water places, and performed worst in water areas.

More stable and systematic performances appeared among the four IMERG estimates for the second level of evaluation with sufficient data supports. Urban and water areas had larger conditional mean rain rates, together with higher RMSE than other places. Compared with water bodies, random errors were more prominent in urban districts, especially for morphed PMW and the mixture of morphed PMW and IR estimates, with larger RMSEs based on similar mean rain rates. For the systematic error, PMW does not demonstrate obvious differences among different surface types, but IR clearly underestimated in urban and non-urban, non-water areas, with the biases of −9.8% and −5.7%. The morphing algorithm brought a similar systematic error to the PMW estimate as IR, resulting in the biases of −6.3% and −11% in urban and non-urban, non-water places, and these biases were further enlarged after the mixing of IR estimate, to −22% and −19%. For the CC indicator, excepting the morphed PMW, which had a lower CC in urban areas than other surfaces, all estimates performed better in urban and non-urban, non-water areas with higher CC value, but suffered from low CC in water areas.

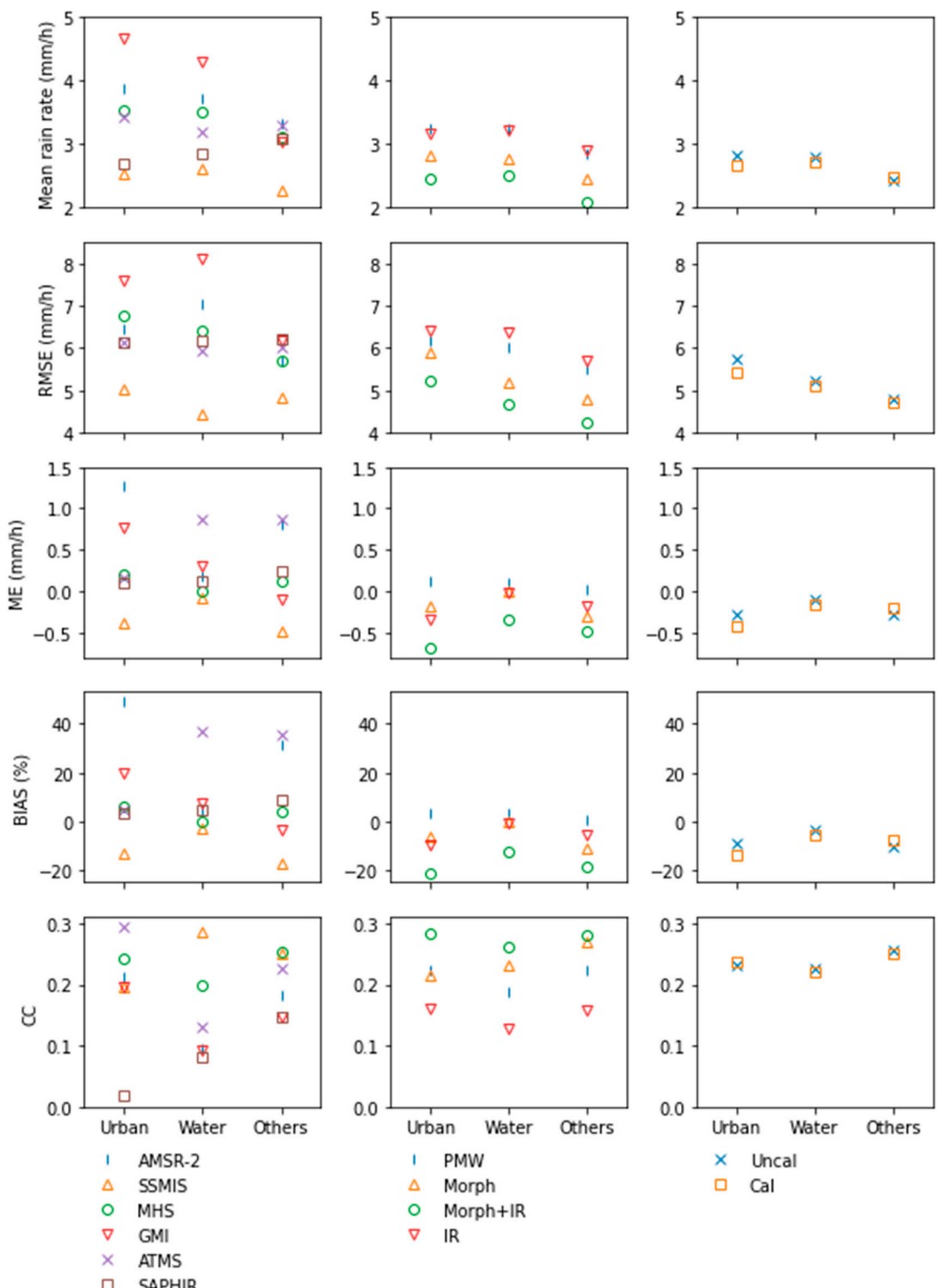

**Figure 9.** Mean value and four continuous indices, RMSE, ME, BIAS, and CC, of the hit data of 12 IMERG estimates in urban, water, and other areas. The left, middle, and right columns indicate the IMERG estimates for the first, second, and third levels of evaluations.

For the third level of evaluation concerning monthly gauge calibration, the differences in the conditional mean rain rate for three categories of surface type were diminished, as smaller in urban and water areas and larger in other places. Random errors were partly removed after monthly gauge calibration with lower RMSE values. The bias, the indicator of systematic errors, in urban and

water areas, was exacerbated from −8.9 to −13.8% and from −3.3 to −5.6%, while in broad non-urban, non-water areas, it improved from −10.3 to −7.4%. As for CC, only the estimate in urban areas improved after the gauge calibration, and the performance in water and other areas declined.

### 3.3.2. Distance to the Coast

The influences of the coast position on IMERG error are studied in this section. Study areas were limited to the zone within 300 km from the coast, since most of the research pixels are in this zone. For the performance assessment of the precipitation detectability, the data sample within 300 km from the coast was divided into 15 groups, with the distance band as 20 km per group.

Figure 10 displays the variation of POD, FAR, and CSI, three categorical indexes according to the distance to the coast for 12 IMERG estimates. The PODs for six PMW instruments declined within 80 km from the sea as the distance rose, among which SSMIS was influenced most by the distance factor with the range of 0.08 from 0.62 to 0.54. Beyond 80 km from the coast, the PODs of two imagers, SSMIS and AMSR-2, did not display obvious trends; both performed better than other sensors in inland districts (>120 km from the sea), while others had declining PODs up to a distance of around 200 km from the sea. For FAR, most PMW sensors had larger fractions of false alarms in more coastal areas than in inland areas, within 80 km of the sea, while ATMS manifested unusually higher FAR in more inland places. With low FAR varying between 0.40 and 0.45, MHS had a stable performance in research areas. Finally, the CSI index gives a glance at the comprehensive performance on precipitation detectability. Three PWM sensors, ATMS, GMI, and MHS, performed better in more coastal areas with higher CSI values, while SSMIS, AMSR-2, and SAPHIR yielded better results in inland places.

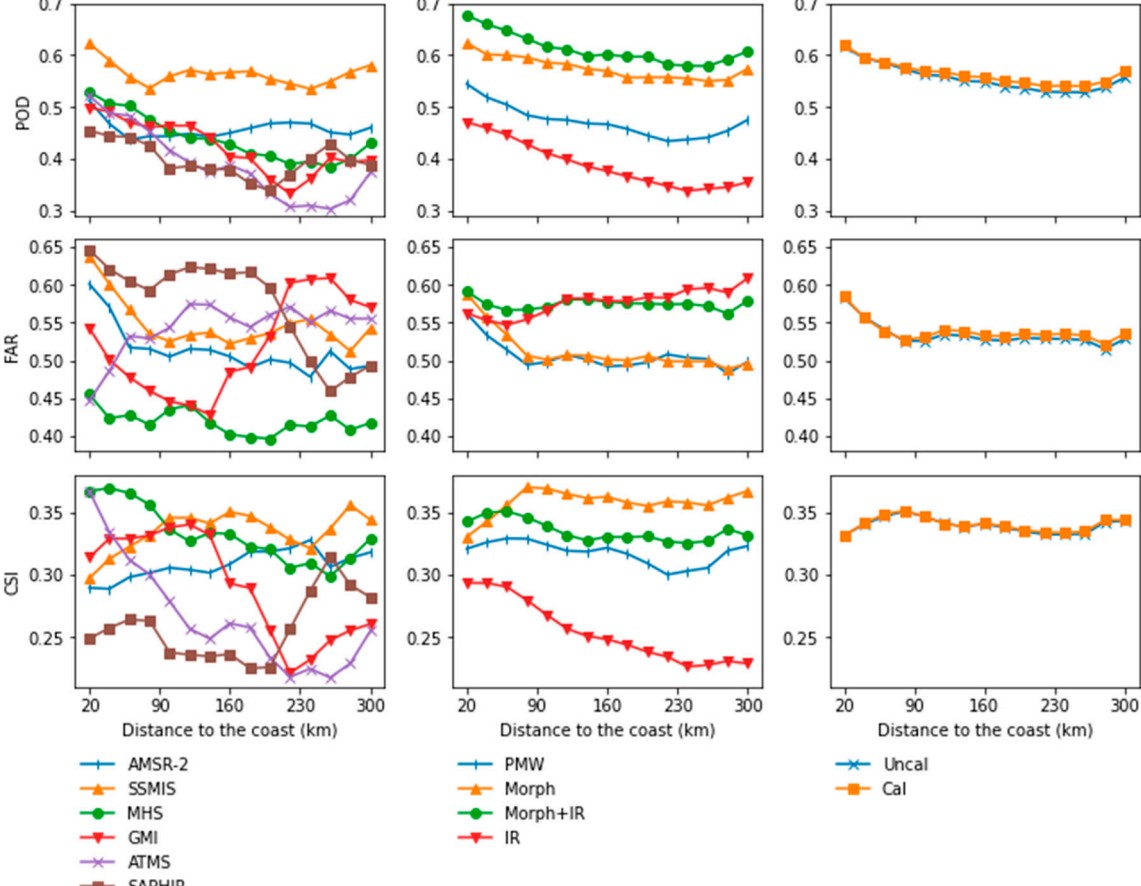

**Figure 10.** Three categorical indices, POD, FAR, and CSI, for 12 IMERG estimates according to the distance to the coast. The left, middle, and right columns indicate the IMERG estimates for the first, second, and third levels of evaluations.

For the second level of assessment, the PODs of four precipitation estimates decreased gradually from coast to inland until about 240 km from the sea, and then increased, among which the distance to the sea had larger impacts on the IR and PMW estimates than on the morphed PMW and the mixture of PMW and IR estimates. Larger ratios of false alarms were detected in coastal areas in the range of 80 km to the sea, and the merged and morphed PMW estimates appeared to be more sensitive to the distance to the coastline than the IR and the mixture of PMW and IR estimates. Further inland, the FAR of IR estimates kept rising, however, which performed stable for the other three estimates. In general, IR had worse performance in inland (with the CSI as 0.23) than coastal areas (with the CSI as 0.29).

The influences of monthly gauge calibration were evaluated in the last level of evaluation. It can be found that the POD index improved in inland areas at the cost of larger FAR, while the CSI remained almost the same.

Then, the quantitative evaluation for the hit events was conducted, divided into six groups, with the band of the distance to the sea as 50 km per group, ensuring enough hit data in each group. Figure 11 shows the conditional mean rain rate and RMSE of the hit IMERG estimates according to different distances to the coast. It is obvious that the RMSE and mean rain rate of IMERG estimate obey linear regression with the slope as 1.9, and the coastal areas suffered from more random errors with higher RMSEs.

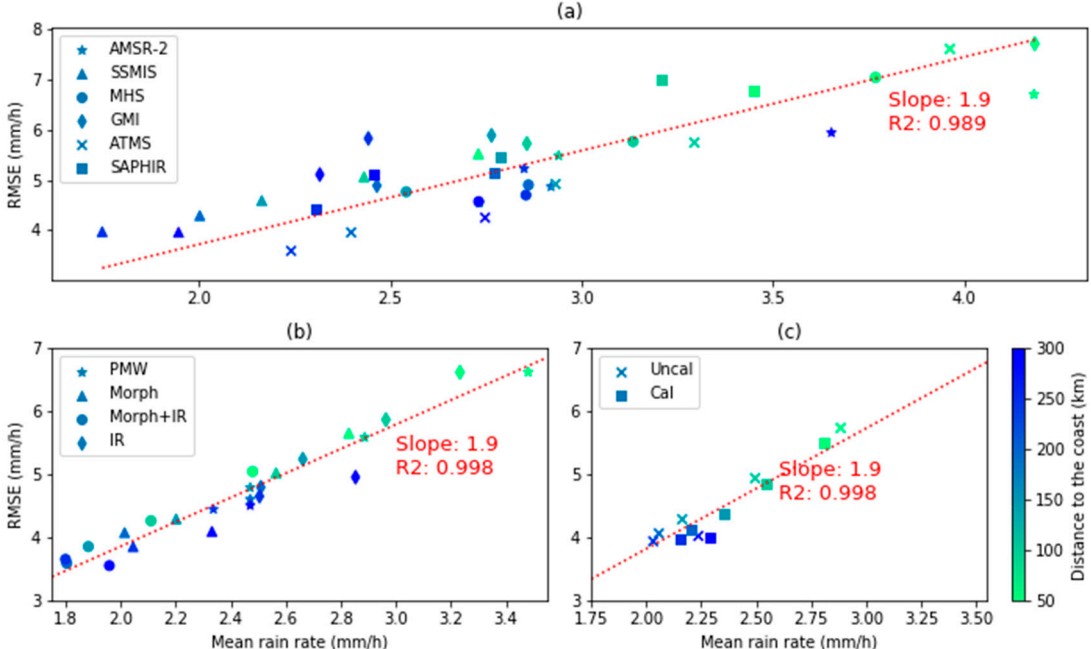

**Figure 11.** The scatter plot of RMSE and mean rain rate of the hit data of 12 IMERG estimates according to different distances to the coast. (**a**) indicates six individual PMW estimates, and (**b**) demonstrates four multi-satellite estimates for the second level of evaluation, among the merged PWM, morphed PWM, the mixture of morphed PMW and IR, and IR-only estimates, and finally (**c**) shows the multi-satellite estimates with and without gauge calibration.

Among six individual PMW sensors in Figure 11a, three instruments, SSMIS, GMI, and SAPHIR displayed obvious random errors with most RMSEs above the regression line, among which GMI suffered from larger random errors in inland (>200 km from the sea) than coastal areas. The last three PMW sensors, AMSR-2, MHS, and ATMS, had relatively small random errors, especially ATMS and MHS in inland districts (>150 km from the sea). Next, the BIAS and CC indices are shown in Figure 12a. It can be seen that coastal areas benefited from higher CC values for most PMW sensors, except for GMI, and the bias varies greatly according to the distance to the coast. SSMIS performed regardless of the distance to the sea, with biases between −9.0% and −27%, while ATMS and AMSR-2 overestimated considerably for six distance groups, especially the most inland group. SAPHIR overestimated in

coastal areas (<200 km to the sea) but underestimated in inland places (>200 km to the sea). The biases of GMI ranged between −16% and 33%. MHS had relatively small systematic errors in coastal areas but overestimated in inland areas (>150 km from the sea) with the biases between 14% and 17%.

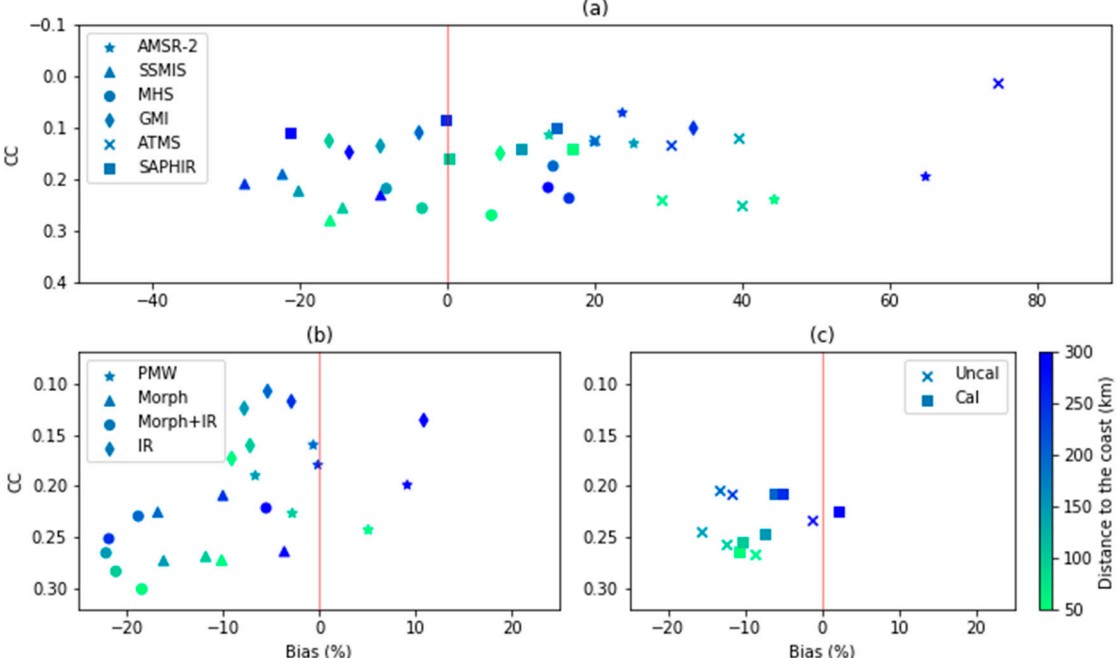

**Figure 12.** The scatter plot of CC and BIAS of the hit data of 12 IMERG estimates according to different distances to the coast. (**a–c**) indicate the IMERG estimates for the first, second, and third levels of evaluations.

For the second level of assessment, coastal areas (<100 km from the sea) had significant random errors, as the RMSEs for four IMERG estimates are above the regression line, as shown in Figure 11b. As for the CC score in Figure 12b, the precipitation estimations in inland places suffered from a smaller CC score than coastal areas. The biases of merged PMW shifted between negative and positive values, while with the overall reduction of mean rain rate for all distance groups, the degree of underestimation grew after the PMW morphing, and PMW and IR mixing. In the areas within 150 km from the sea, there was less underestimation in more coastal areas for three estimates, except for the IR-only.

Comparing between the last two estimates, the monthly gauge calibration reduced the differences between the inland and coastal areas and adjusted the spatial distribution of precipitation as larger condition rain rates were detected in inland areas (>50 km from the sea) and lower rates for coastal districts (<50 km from the sea) as shown in Figure 11c, and the gauge calibration brought more stable performance for IMERG product according to the distance to the sea. The systematic underestimation was effectively corrected after gauge calibration with a lower absolute value of bias, especially the inland areas (>100 km from the sea), while there was no significant change on the CC index.

## 4. Discussion

For the first level of evaluation, six PMW sensors have distinct performances relating to on the specific operation region and time. Chen et al. analyzed the source errors of GPM-GSMaP over mainland China from April 2014 to March 2018 and found that imagers are superior to sounders [23], but in this research, no significant differences between imagers and sounders is shown. Moreover, Chen et al. reported that SSMIS performs well with lower total errors, while GMI overestimates greatly. However, SSMIS demonstrates large random errors, while GMI is generally reliable in the IMERG study of Tan et al. in the mid-Atlantic region of the United States [10]. In this research, SSMIS performs well on precipitation detectability, which might be due to sensitivity to precipitation in

high frequency channels [27], and GMI stands out from other PMW sensors with high detectability over water bodies. Based on the distinctive characteristics of each PMW sensors, except for the algorithm update of precipitation retrieval from PMW satellites, the improvement of the merged PMW estimate can exploit a conditional merging method with different trust weights for each sensor, according to their performance in each landscape, to assimilate a high quality merged PMW product. Because of the limitation of ground data, this research is limited to the southeast coastal region of China in the rainy season of 2018, but larger spatiotemporal research with finer time resolution will be needed to advance our understanding of the typical features of each PMW sensors in different landscapes.

For the second level of assessment, among the merged and morphed PMW, the mixture of PMW and IR, and IR-only estimates, PMW estimates have better overall performance than IR, which is similar to the results of Tan et al. [10]. Besides, this study identified the heterogeneous errors in PMW and IR estimates, as PMW tends to overestimate light (<1 mm/h) and extreme large rain rates (>50 mm/h) and underestimates medium rain (around 20 mm/h), while IR aberrantly caps the extreme large rain events within 54 mm/h. Overlooking the heterogeneous errors within estimates, the morphing (PMW) and combing (PMW and IR) algorithms improve the precipitation estimate as a whole at the sacrifice of the deterioration of part of observations or performance indicators. In this study, the mixture of PMW and IR estimate had higher CC (0.28) than the merged PMW (0.22) by correcting the overestimation on extreme rain events, while the bias is enlarged from 1.0 to −18.7%. Cluster morphing and combing methods can be developed to solve this problem by processing each type of data individually according to different precipitation intensities or elevation and climate conditions, to avoid or alleviate the error propagation from a certain type of data, and hence increase the reliability of the whole product.

## 5. Conclusions

In this research, three intermediate precipitation estimates, as well as the final product within IMERG version 6, are partitioned into 12 categories of estimates which are validated against the CMPA product in three levels of evaluation (a. six individual PMW; b. merged PMW, morphed PMW, the mixture of PMW and IR, IR-only, and c. multi-satellite estimates without and with gauge calibration) over southeast coastal areas of China, and the influences of topography and surface type factors on these error sources are identified. The three research objectives concerning the general evaluation and the influences of topography and surface type are concluded as follows:

1.  In general, MHS has superior and stable comprehensive behavior among six individual PMWs. In general, MHS has outstanding and stable comprehensive behavior among six individual PMW sensors, and SSMIS operates well in terms of precipitation detectability, while SAPHIR has the worst performance on both precipitation detectability and quantitative estimation. The merged PMW estimate has the POD and FAR of 0.48 and 0.52, respectively, and tends to overestimate both light (<1 mm/h) and extremely large rain rates (>50 mm/h) and underestimates medium rain (around 20 mm/h). IR has undesirable overall performance with the POD and FAR of 0.40 and 0.57, respectively, and aberrantly caps all extreme rain events within 54 mm/h. The biases of merged PMW and IR are 1.0% and −5.7%. The morphed PMW and the mixture of PMW and IR estimates detect more precipitation events, increasing the POD and FAR to 0.62 and 0.58, while the conditional mean rain rate is decreased, leading to significant underestimation, with a bias of −18%. After the monthly gauge calibration, most of the indicators slightly improve.

2.  For the topographic influence, more precipitation events are detected in lower places with larger condition mean rate rates than highlands, and therefore suffer from larger random errors. In the first level of evaluation among six PMW sensors, except for MHS and SSMIS, the biases for the other four PMW instruments are sensitive to the elevation change and vary between 40% and −20%. IR estimate displays worse precipitation detectability in highlands with lower CSI which is stable for PMW estimates. Besides, PMW estimates have larger CC in high elevations, which characteristic further propagate to the final estimate. The monthly gauge calibration mitigates the elevation impact on the errors.

3.  More precipitation with larger quantitative uncertainty is recognized in urban and water body areas, while other places have a stable performance with higher CSI scores. Different from other PMW sensors, GMI shows good precipitation detectability over water areas. The gauge calibration shrinks the differences among urban, water, and other places. As for the distance to the coast, coastal areas have more precipitation than inland places with larger POD and FAR. ATMS, GMI, and MHS have better detectability over coastal areas with higher CSI values, while SSMIS, AMSR-2, and SAPHIR yield better results in inland places. For six estimates in the second and third levels of evaluations, within 240 km from the sea, the POD indexes decrease gradually from coast to inland districts, whose range for FAR is 80 km from the sea. The CSI of IR decreases from 0.29 for the most coastal group to 0.23 for the most inland places. The conditional RMSE and mean rain rate of 12 IMERG estimates obey a linear regression with the slope as 1.9, according to different distances to the coast. The monthly gauge calibration reduces the differences between the inland and coastal areas and adjusts the spatial distribution of precipitation as larger condition rain rates are detected in inland areas (>50 km from the sea) and less for coastal districts (<50 km from the sea).

**Author Contributions:** Conceptualization, X.S.; Data curation, Z.L., Z.M., J.X. and S.Z.; Formal analysis, X.S.; Funding acquisition, Z.M.; Methodology, X.S. and Z.L.; Project administration, Z.M.; Supervision, Z.M.; Writing—original draft, X.S.; Writing—review & editing, Z.L., Z.M. and H.L. All authors have read and agreed to the published version of the manuscript.

**Funding:** This study was financially supported by the National Natural Science Foundation of China (grant no. 41901343); the Key R&D Program of the Ministry of Science and Technology, China (grant no. 2018YFC1506500); the Second Tibetan Plateau Scientific Expedition and Research (STEP) program (grant no. 2019QZKK0105); and the Open Fund of the State Key Laboratory of Remote Sensing Science, China (grant no. OFSLRSS201909).

**Conflicts of Interest:** The authors declare no conflict of interest.

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
