# Peer review of "Ground Validation and Error Sources Identification for GPM IMERG Product over the Southeast Coastal Regions of China"

_remotesensing, doi:10.3390/rs12244154_

Round 1

Reviewer 1 Report

This manuscript investigates the error sources of the latest IMERG version 6, an algorithm for remote sensing-based global precipitation measurement, using China Merge Precipitation Analysis hourly V1.0 product (CMPA) as a reference data. While I think this research is well designed, I would like to suggest the authors consider the following points to make the evaluation more reliable.

1) Because the evaluation relies on the CMPA product, the accuracy of CMPA needs to be described in more detail. The author said that there is a high-density gauge-based precipitation observation in southeast China, but are rain gauges distributed evenly in the study area? I expect to see a map of rain gauges.

2) The maps of land cover/surface and distance from the coast should be added.

3) In my opinion, the statistical metrics provide limited information because they are only average errors or deviations between predicted and reference values. The error might vary a lot depending on location. Estimating spatial variation in error between IMERG and CMPA would be much more meaningful. Please refer to this paper, https://doi.org/10.1016/j.jag.2018.09.020, for how to estimate spatial distributions of statistical metrics.

Minor revisions:
- The acronyms, such as GPM, PMW, etc., need to be explained the first time they appear.
- I cannot understand the meaning of the "amount of data" column in Table 2. Please scientifically write numbers (e.g., 12,345, not 12345).

Author Response

Thanks.

Reviewer 2 Report

c1: Abstract: Please explain the acronyms at their first appearance in the text!

c:2 The description of the Methodology should be enriched. You should provide more details regarding the workflow and the software used in this regard.

c3: Please explain the terms of the equations included in Table 3 by adapting them to your study.

c4: The Discussion section should be extended! You should compare the results from your study with the results achieved in other similar studies. 

Author Response

Thanks.

Round 2

Reviewer 1 Report

I think the manuscript is improved, and the authors made acceptable responses in resolving my concerns.

Reviewer 2 Report

Dear authors, 

Thank you for taking into account my comments in order to improve your paper! Now, the paper can be accepted in this form!